statistics

COVID-19, SARS-CoV-2, reverse transcription–polymerase chain reaction, chest computed tomography, serological tests, multiple tests integration

**Author for correspondence:**
Maggie H. Wang
e-mail: maggiew@cuhk.edu.hk

†Joint first authors.

# A Bayesian method for synthesizing multiple diagnostic outcomes of COVID-19 tests

Lirong Cao[1,4,†], Shi Zhao[1,4,†], Qi Li[1,4,†], Lowell Ling[2], William K. K. Wu[2], Lin Zhang[2], Jingzhi Lou[1], Marc K. C. Chong[1,4], Zigui Chen[3], Eliza L. Y. Wong[1], Benny C. Y. Zee[1,4], Matthew T. V. Chan[2], Paul K. S. Chan[3] and Maggie H. Wang[1,4]

[1]JC School of Public Health and Primary Care, [2]Department of Anaesthesia and Intensive Care, [3]Department of Microbiology, Stanley Ho Centre for Emerging Infectious Diseases, Li Ka Shing Institute of Health Sciences, The Chinese University of Hong Kong, Hong Kong
[4]Clinical Trials and Biostatistics Lab, CUHK Shenzhen Research Institute, Shenzhen, People's Republic of China

LC, 0000-0002-5793-3878; SZ, 0000-0001-8722-6149; MKCC, 0000-0001-5610-1298; MHW, 0000-0003-1223-4595

The novel coronavirus disease 2019 (COVID-19) has spread worldwide and threatened human life. Diagnosis is crucial to contain the spread of SARS-CoV-2 infections and save lives. Diagnostic tests for COVID-19 have varying sensitivity and specificity, and the false-negative results would have substantial consequences to patient treatment and pandemic control. To detect all suspected infections, multiple testing is widely used. However, it may be challenging to build an assertion when the testing results are inconsistent. Considering the situation where there is more than one diagnostic outcome for each subject, we proposed a Bayesian probabilistic framework based on the sensitivity and specificity of each diagnostic method to synthesize a posterior probability of being infected by SARS-CoV-2. We demonstrated that the synthesized posterior outcome outperformed each individual testing outcome. A user-friendly web application was developed to implement our analytic framework with free access via http://www2.ccrb.cuhk.edu.hk/statgene/COVID_19/. The web application enables the real-time display of the integrated outcome incorporating two or more tests and calculated based on Bayesian posterior probability. A simulation-based assessment demonstrated higher accuracy and precision of the Bayesian probabilistic model compared with a

single-test outcome. The online tool developed in this study can assist physicians in making clinical evaluations by effectively integrating multiple COVID-19 tests.

# 1. Background

The ongoing coronavirus disease 2019 (COVID-19) pandemic is causing substantial morbidity and mortality globally [1]. Diagnosis is of importance to control outbreaks, especially in the absence of specific treatments or vaccines. To date, COVID-19 is commonly diagnosed by the detection of unique sequences of SARS-CoV-2 RNA using the nucleic acid amplification tests (NAAT), e.g. real-time reverse transcription–polymerase chain reaction (RT-PCR) [2]. However, suboptimal sample collection, storage and transportation may lead to false-negative results. The sensitivity of laboratory-based molecular testing is largely dependent on the types of specimens and the time of collection from the onset of illness [3], which also leads to an extensive range of sensitivity for RT-PCR in previous studies, between 46% and 92% [4–8].

As the RT-PCR test might fall short of testing capacity and timeliness in some regions, recent studies proposed that chest computed tomography (CT) scans could be included as a supplementary diagnostic tool if there are clinical symptoms, epidemiological characteristics and imaging characteristics of viral pneumonia that are compatible with COVID-19 infection in epidemic areas [8]. Although using chest CT may be helpful, the specificity is low due to the absence of pathognomonic CT features, which is even lower than 50% according to earlier research findings [9]. In addition, serological tests are also recommended as a supplement for nucleic acid detection, because the antibody-based methods are relatively cheap, easy to operate and have lower technical requirements [10]. Since the detection of antibodies against SARS-CoV-2 is more accurate in the middle to later stages of COVID-19, antibody tests are primarily used to determine whether a person has been previously infected. In many prior reports, the sensitivity for combined IgM and IgG detection is higher than 71% and the specificity is higher than 90% [11–16].

None of the commonly used diagnostic tests, e.g. RT-PCR, chest CT and serological tests, alone is sufficiently accurate to provide absolute diagnostic certainty [10,17]. In view of the advantages and shortages of each detection method, parallel or serial tests are recommended in the clinic and the results are cross-referenced to improve diagnostic yield [18]. However, clinicians will face immense difficulty diagnosing COVID-19 when the test results are inconsistent. In this study, we provided a Bayesian method to synthesize multiple diagnostic outcomes and developed an online tool to evaluate the probability of an individual being infected by SARS-CoV-2.

# 2. Methods

## 2.1. Bayesian probabilistic framework

With the knowledge of the sensitivity and specificity of each COVID-19 diagnostic test, we constructed a Bayesian probabilistic model to synthesize multiple testing outcomes for an individual subject. We calculated the posterior probability of having COVID-19 based on the information and the outcomes of more than one test. Although the outcomes of different tests may be correlated, they are assumed conditionally independent with the disease status fixed when these tests are based on the different biological attributes [19]. Thus, the Bayesian probabilistic framework is applicable to infer the probability of disease status with correlated testing outcomes [20–22].

We considered that one individual subject receives $M$ diagnostic tests, where $M$ is an integer and $M > 1$. The testing outcome, i.e. positive or negative, is denoted by $T_i$ for the $i$th test. We defined $T_i$ as a binary outcome that is 1 for positive testing outcome and 0 otherwise. We denoted the event that 'the individual subject has COVID-19' by $D$ (stands for 'diagnosed' or 'diseased'). For convenience, we also denoted the complement of event $D$, i.e. 'the individual subject does not have COVID-19', by $N$ (stands for 'not diagnosed'). Straightforwardly, the summation of the probabilities (**Pr**) of $D$ and $N$ was 1.

The posterior probability of $D$ conditional on the $M$ testing outcomes is $\mathbf{Pr}(D \,|\, T_1, T_2, \ldots, T_M)$. Hence, by using the Bayes theorem, the $\mathbf{Pr}(D \,|\, T_1, T_2, \ldots T_M)$ can be computed by using equation (2.1). Based on the intuition of the Bayes framework, the test with higher sensitivity or specificity will be automatically

**setup testing information and outcomes** (help)(example)

| test #1: RT-PCR test | included ∨ | sensitivity (#1) | 0.75 | specificity (#1) | 0.95 | outcome (#1) | positive ∨ |
| test #2: chest CT test | included ∨ | sensitivity (#2) | 0.94 | specificity (#2) | 0.37 | outcome (#2) | positive ∨ |
| test #3: antibody test | included ∨ | sensitivity (#3) | 0.65 | specificity (#3) | 0.98 | outcome (#3) | positive ∨ |

**setup pre-test probability for someone with suspected exposure**

pre-test probability    0.5

**results**

you have selected 3 test(s)

you have set the pre-test probability at 0.5 in the subject population.

the posterior probability of the subject being infected with COVID-19 is 0.999.

**Figure 1.** User interface for COVID-19 diagnostic assessment tool.

assigned with a higher weight,

$$
\begin{aligned}
\mathbf{Pr}(D|T_1, T_2, \ldots, T_M) &= \frac{\mathbf{Pr}(T_1, T_2, \ldots, T_M|D) \cdot \mathbf{Pr}(D)}{\mathbf{Pr}(T_1, T_2, \ldots, T_M)} \\
&= \frac{\mathbf{Pr}(T_1, T_2, \ldots, T_M|D) \cdot \mathbf{Pr}(D)}{\mathbf{Pr}(T_1, T_2, \ldots, T_M|D) \cdot \mathbf{Pr}(D) + \mathbf{Pr}(T_1, T_2, \ldots, T_M|N) \cdot \mathbf{Pr}(N)} \\
&= \frac{\mathbf{Pr}(D) \cdot \prod \mathbf{Pr}(T_i|D)}{\mathbf{Pr}(D) \cdot \prod \mathbf{Pr}(T_i|D) + \mathbf{Pr}(N) \cdot \prod \mathbf{Pr}(T_i|N)}.
\end{aligned}
\tag{2.1}
$$

Here, $\mathbf{Pr}(T_i = 1 | D)$ is the sensitivity of the $i$th test, and $\mathbf{Pr}(T_i = 0 | D)$ is its $(1 - \text{sensitivity})$. The $\mathbf{Pr}(T_i = 0 | N)$ is the specificity of the $i$th test, and $\mathbf{Pr}(T_i = 1 | N)$ is its $(1 - \text{specificity})$. The $\mathbf{Pr}(D)$ indicated the pre-test probability of having COVID-19 for an individual who receives tests. An alternative interpretation of $\mathbf{Pr}(D)$ is the prevalence of COVID-19 among the testing subjects if there is no contact history or symptom. The prevalence among testing populations is available for each region in the 'Our World in Data' website [23]. The testing performance of each COVID-19 diagnostic test can be summarized from existing literature or obtained from the clinical evaluations of commercial testing kits. Hence, $\mathbf{Pr}(D | T_1, T_2, \ldots, T_M)$ is computable. Additionally, if the testing subjects are at higher exposure risk, like healthcare practitioners or customs officers, the pre-test probability should be adjusted to a higher range and vice versa.

## 2.2. COVID-19 diagnostic assessment tool

The theoretical framework can be conducted to integrate multiple diagnostic test results for COVID-19. To facilitate public use, we set up an open-access and user-friendly online application of our framework. The application is available at http://www2.ccrb.cuhk.edu.hk/statgene/COVID_19/. The user interface was designed to be intuitive, see figure 1. The 'setup testing information and outcomes' section offers users an option to indicate conducted tests, sensitivity and specificity, and testing outcome (positive or negative). Test 1, Test 2 and Test 3 can be diagnostic tests such as RT-PCR, chest CT and antibody test, or they can be repeated measures of a same test. After inputting the pre-test probability of an individual with suspected exposure, the 'results' section will show the posterior probability of the subject being infected with COVID-19 according to the above settings. A detailed user manual is also available in the web application. Many factors, such as variation of the incubation period, the severity of disease and sample quality, might impact the sensitivity and specificity of diagnostic tests [24]. Users are suggested to adjust these parameters accordingly.

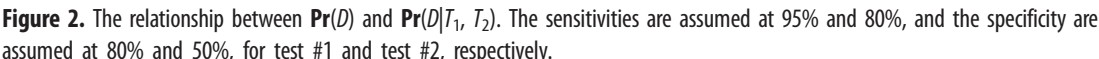

**Figure 2.** The relationship between **Pr**(D) and **Pr**(D|$T_1$, $T_2$). The sensitivities are assumed at 95% and 80%, and the specificity are assumed at 80% and 50%, for test #1 and test #2, respectively.

## 2.3. Performance evaluation for the Bayesian method

We further simulated disease diagnosis according to the posterior probability. The **Pr**($D | T_1, T_2, \ldots, T_M$) > 0.5 was considered as positive cases for COVID-19. The tested population set in the simulation modelling was set as one million. The performance of the combined method by integrating multiple testing outcomes was compared with that of a single test by accuracy and precision. The accuracy and precision can be, respectively, computed in equations (2.2) and (2.3).

$$\text{Accuracy} = \frac{(\text{TP} + \text{TN})}{(\text{TP} + \text{TN} + \text{FP} + \text{FN})} \tag{2.2}$$

and

$$\text{Precision} = \frac{\text{TP}}{(\text{TP} + \text{FP})}. \tag{2.3}$$

Here, TP is the number of true positive subjects, FP is the number of false-positive subjects, TN is the number of true negative subjects and FN is the number of false-negative subjects. All analysis was conducted in **R** statistical software (v. 3.5.1) [25].

## 3. Results

In figure 2, we demonstrated the relationship between the probability of having COVID-19 and the prevalence of COVID-19 among testing subjects under a two-test scenario as an example. The sensitivities were assumed to be at 95% and 80%, and the specificities were assumed to be at 80% and 50%, for test #1 with testing outcome $T_1$ and test #2 with testing outcome $T_2$, respectively. Four combinations of the testing outcomes were considered including ($T_1$ = positive, $T_2$ = positive) in figure 2a, ($T_1$ = positive, $T_2$ = negative) in figure 2b, ($T_1$ = negative, $T_2$ = positive) in figure 2c, and ($T_1$ = negative, $T_2$ = negative) in figure 2d. The online tool is also available for three diagnostic tests synthetization and serial testing synthetization (http://www2.ccrb.cuhk.edu.hk/statgene/COVID_19/).

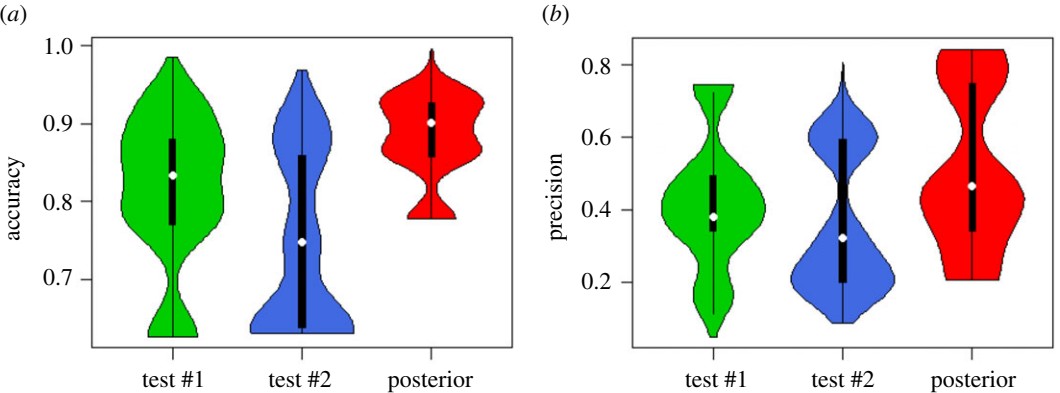

**Figure 3.** Violin plot of the accuracy and precision for test #1, test #2 and the Bayesian probabilistic model. The pre-test probability is assumed from 0.001% to 25%, and sensitivities and specificities for both T1 and T2 are assumed from 55% to 100%.

For comparison purposes, we reported the performance of a single test and the Bayesian probabilistic model for multiple tests in COVID-19 diagnosis, given a pre-test probability from 0.001% to 25%, sensitivities and specificities for both $T_1$ and $T_2$ from 55% to 100% in figure 3. $T_1$ and $T_2$ could be two types of tests, or repeated tests for different samples. The mean accuracy for the Bayesian probabilistic model is 88%, which is higher than 81% for $T_1$ and 74% for $T_2$. The mean precision is also largest for the Bayesian probabilistic model (51%) compared with $T_1$ (43%) and $T_2$ (36%) alone. We also performed another simulation by setting a lower pre-test probability (0.001–5%), see electronic supplementary material, figure S1. The results are consistent, and both show that the performance of the Bayesian model is much better than individual tests.

# 4. Discussion

Rapid and accurate diagnosis of SARS-CoV-2 infections could facilitate timely control of the outbreaks of COVID-19 by early detection. The characteristics of three detection methods, RT-PCR, chest CT and serological tests, against COVID-19 are distinct, which may cause large variation in accuracy when these tests are applied to different courses of the illness or patients with non-identical symptoms [26]. The testing performance for RT-PCR against different types of specimens varies according to the previous studies. Outcomes obtained from samples of the lower respiratory tract, like sputum specimens, are more accurate than that of the upper tract, e.g. nasal swabs and throat swabs, in COVID-19 diagnosis [27,28]. Since the exact time of onset is sometimes unknown, particularly for mild patients, and not all patients with COVID-19 could produce sputum for diagnostic evaluation, parallel or serial testing is widely used in COVID-19 diagnosis. However, clinicians may face uncertainty in the disease diagnosis when one test is positive but the other is negative. We developed a probabilistic model and a web application to synthesize the risks of having COVID-19 considering three determining factors that included test sensitivity, specificity and pre-test probability. In serial diagnostic tests applied on a same subject, inconsistent testing results might be observed as a result of the change of underlying disease status rather than the power of the tests themselves. For example, a patient that tested negative by PCR when first tested shortly after exposure, may not yet have detectable virus, but could later test positive once viral loads are sufficient incubated. In application, clinicians would also need to consider factors such as exposure risk before and between tests to interpret results. Although the test results from the same subject are highly correlated, they are conditionally independent, and thus the correlation will not affect the calculation of the posterior probability. The second advantage for the Bayesian model is that suitable weights will be added to the corresponding diagnostic tests according to their sensitivities and specificities, especially when the weight for each test is hard to artificially determine. Multiple tests combined with our diagnostic tool demonstrated improved diagnostic accuracy compared with individual tests.

To contain the outbreak of COVID-19, rapid and accurate diagnostic tests are critical. RT-PCR detects the presence of the specific genetic material with generally high specificity but limited sensitivity. The chest CT method, a valuable tool for the triage of suspected cases, is sensitive but prone to high false-positive. A recent study reported that some patients were test-negative by RT-PCR, but radiological evidence detected lung lesions compatible with COVID-19 [29]. The specificity of chest CT was

inconsistent in prior studies [30], which may be caused by the differences in study design of independent investigations and the differences in diagnostic experiences of thoracic radiologists. For antibody-based methods, the sensitivity and specificity are higher for combined IgM and IgG detection compared with using any of antibodies alone. However, antibodies were reported to appear after the onset of symptoms [26]. The presence of IgG is particularly delayed. Therefore, serological tests are more sensitive to detect infections after 7 days of symptom onset [31]. A negative serological test conducted in the early stage of disease onset may not be sufficient to rule out suspected cases. Comprehensive integration of multiple testing results, especially results from the rapid diagnostic tests with lower sensitivity and specificity, can provide a more accurate result.

During the surging phase of pandemic, the availability of RT-PCR is often affected by a shortage of laboratory test kits, long waiting time, complex operation, expensive equipment and lack of specialized technicians. Alternative testing methods might be applied to assist the diagnosis of COVID-19. Chest CT as a routine imaging method is often readily accessible in general hospitals. Serological tests, being fast and simple to perform, are also widely accepted in clinical and public health settings. High sensitivity is important for screening and diagnosing COVID-19, considering that asymptomatic cases also have a risk of spreading the virus, while specificity needs to be considered to avoid false treatment or nosocomial infection. Using the Bayesian probabilistic model could fully use the outcome of each detection method or testing sample and showed improved performance in COVID-19 diagnosis.

The main limitation of this study is that due to the lack of observational research data, the performance of the Bayesian method can only be evaluated through simulation models and pre-set cut-off values for positive diagnoses. However, the calculation of the posterior probability estimation will not be affected. The only impact caused by this inevitable limitation is that the absolute improvement for the performance of the Bayesian framework will be different in the real world.

The decision analytical model found that the probability of having COVID-19 calculated based on multiple testing varied with pre-test probabilities, which would be an important factor to consider, particularly in regions with increasing disease prevalence. Pre-test probability, an estimate of a person's chance of being infected before testing, mainly depends on the local positive rate of COVID-19 among testing samples, SARS-CoV-2 exposure history and clinical signs [32]. Someone who is feeling unwell after close contact with suspected patients may have a higher risk of COVID-19 compared with local prevalence. The online tool developed in this study can assist physicians in performing a quick diagnosis onsite. The statistical underpinning of the proposed method is generic and can be applied to other infectious diseases.

# 5. Conclusion

Diagnostic tests for COVID-19 currently in use have varying sensitivity and specificity. As parallel and serial testing are widely used in the clinic to avoid missed diagnosis or misdiagnosis, we presented a Bayesian probabilistic framework that takes advantage of the outcome of multiple detection methods or testing samples for detecting COVID-19. We also developed a convenient online tool to display the posterior probability of being infected by SARS-CoV-2 by effectively integrating multiple COVID-19 tests. The online tool adds value to existing detection methods and can aid physicians in making clinical evaluations.

Ethics. Neither ethical approval nor individual consent is applicable.

Data accessibility. No real-world data are used in this study. The code is provided in the electronic supplementary material [33].

Authors' contributions. M.H.W. conceived the study. L.C. conducted the literature review. S.Z. and M.H.W. carried out the analysis. L.C., S.Z., Q.L. and M.H.W. discussed the results. L.C. and S.Z. drafted the first manuscript. S.Z. and Q.L. developed the online tool. L.L., W.K.K.W., L.Z., J.L., M.K.C.C., Z.C., E.L.Y.W., B.C.Y.Z., M.T.V.C. and P.K.S.C. critically read and revised the manuscript and gave final approval for publication.

Competing interests. M.H.W. is a shareholder of Beth Bioinformatics Co., Ltd. B.C.Y.Z. is a shareholder of Beth Bioinformatics Co., Ltd and Health View Bioanalytics Ltd. Other authors declared no competing interests.

Funding. This work was supported by the National Natural Science Foundation of China (grant nos. 31871340, 71974165), Health and Medical Research Fund, the Food and Health Bureau, the Government of the Hong Kong Special Administrative Region (grant nos. COVID190103, INF-CUHK-1), and the Chinese University of Hong Kong (grant nos. PIEF/Ph2/COVID/06, 4054600).

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
