## [Peer Review File · Royal Society Open Science]

Review History

RSOS-201867.R0 (Original submission)

Review form: Reviewer 1

Is the manuscript scientifically sound in its present form?

No

Are the interpretations and conclusions justified by the results?

No

Is the language acceptable?

Yes

Do you have any ethical concerns with this paper?

No

Have you any concerns about statistical analyses in this paper?

Yes

Recommendation?

Major revision is needed (please make suggestions in comments)

Comments to the Author(s)

Please see uploaded pdf for specific comments (Appendix A).

My key points are:

Firstly, you make a comparison between your method and the use of a single test to make a diagnosis. I do not think this is a fair comparison to make as it confounds two things – firstly the use of multiple tests to make a diagnosis, which even using a completely naïve method would be expected to improve overall diagnostic accuracy, and the benefit of using your method of weighting each test in a probabilistic framework according to its accuracy. What a ‘naïve’ clinician would usually do would be to look at the outcomes of the multiple tests that they have taken and make a decision based on the proportion of positive tests among those. Again, using a cut-off of 0.5 to determine a positive diagnosis, I think that this would be a much fairer comparison to make with your method to determine its overall benefit.

My second comment is that I think your simulation study is much too restrictive as it is. In the literature, there is usually a great deal of uncertainty as to the exact values of specificity and sensitivity, with ranges of values often given. I think further simulations where you look at the impact of this potential misspecification on your results would make for a better analysis. I think you also need to discuss further the impact of what happens if the underlying disease status of the patient changes across tests. It seems realistic to assume that a patient testing negative on an initial test when not infectious, could then go onto to become infectious by the time of a later test. There is some alluding to this in the discussion but no mention of the impact of this on your results.

I think you also need to justify your comment about conditional independence between tests. Li and Liu (2019) showed this assumption is not valid for tests on the same biological attribute.

I tried to access your online tool, however the page wouldn’t load. I would need to be able to see these as it is a significant part of what your paper is proposing as novel.

Many previous studies have conducted Bayesian analysis of multiple tests and I think a greater reference to this literature and how your method is distinct would emphasise its novelty (e.g. Umemneku Chikere et al (2019); Berkvens et al (2006); Dendukuri (2004)). Is there a methodological novelty or is the novelty that you are applying this method to COVID-19?

References:

Berkvens, Dirk*; Speybroeck, Niko*; Praet, Nicolas*; Adel, Amel†; Lesaffre, Emmanuel‡
 Estimating Disease Prevalence in a Bayesian Framework Using Probabilistic Constraints,
Epidemiology: March 2006 - Volume 17 - Issue 2 - p 145-153
 doi: 10.1097/01.ede.0000198422.64801.8d

Dendukuri, N. and Joseph, L. (2001), Bayesian Approaches to Modeling the Conditional Dependence Between Multiple Diagnostic Tests. *Biometrics*, 57: 158-167.
<https://doi.org/10.1111/j.0006-341X.2001.00158.x>

Li, T., and Liu, P. (2019). Comparison of two Bayesian methods in evaluation of the absence of the gold standard diagnostic tests. *Biomed. Res. Int.* 2019:1374748. doi: 10.1155/2019/1374748

Umehneku Chikere CM, Wilson K, Graziadio S, Vale L, Allen AJ (2019) Diagnostic test evaluation methodology: A systematic review of methods employed to evaluate diagnostic tests in the absence of gold standard – An update. PLOS ONE 14(10): e0223832. <https://doi.org/10.1371/journal.pone.0223832>

Review form: Reviewer 2

Is the manuscript scientifically sound in its present form?

No

Are the interpretations and conclusions justified by the results?

No

Is the language acceptable?

No

Do you have any ethical concerns with this paper?

No

Have you any concerns about statistical analyses in this paper?

No

Recommendation?

Major revision is needed (please make suggestions in comments)

Comments to the Author(s)

The authors have used conditional probabilities to estimate the probability that an individual is infected given the results from multiple tests, accounting for the sensitivity and specificity of each test, and the pre-test probability of infection.

The authors have pitched their method as a means to improve diagnostic accuracy to aid with the control of outbreaks, particularly in the context of COVID-19. They suggest that clinicians could use the associated web tool to provide a more accurate estimate of infection status to patients.

There are some disconnects between the theory – which is sound – and how it might be implemented and utilised in practice:

1. The narrative refers to the need for “timely diagnosis” to contain outbreaks, though the example throughout refers to serological testing as one of the three possible tests. The authors themselves note that serological tests provide an accurate indication of prior infection at later stages of infection, which is not consistent with “timely” diagnosis. Particularly for a pathogen such as SARS-CoV-2, where a substantial portion of transmission is pre-symptomatic. Similarly, CT scans are unlikely to be routine for all potential or suspected cases, but rather just for those with more severe disease and/or requiring hospitalisation. As noted further below, the sensitivity and specificity of these tests varies over time, and each would likely be conducted at different times during the infectious profile of an individual (e.g., PCR early, serology later), making them difficult to be combined simply without additional work to determine the sensitivity/specificity specific to the timing of each test. This makes it difficult to see a situation where these different tests are routinely being performed for timely diagnosis of suspected cases, or the outcomes could be simply and reliably combined. Perhaps it would be appropriate to pitch

the narrative along the lines of using multiple rapid diagnostic tests, each of which have lower sensitivity and specificity, but results of which could be combined for a more accurate result.

2. The method requires specification of a pre-test probability to calculate the probability of being infected given the various test results. The authors propose that clinicians specify this probability to calculate the probability the individual is/was infected. While public data is available on daily incidence of cases, it is not immediately clear how the current prevalence would simply inform this probability (e.g., should this be weighted by the generation interval?). Further, these kinds of high-level data cannot account for individual heterogeneities in risk (e.g., healthcare practitioners treating COVID-19 patients would have a higher pre-test probability than another individual in the same population), and thus how much to adjust the pre-test probability in different contexts. Figure 2 in the submitted manuscript highlights the sensitivity of the outcome to this pre-test probability, particularly in the scenarios where it would be most useful – i.e., where the multiple tests give disparate results – and so accurately choosing this probability is key, but difficult. I would propose that were this approach to be implemented in some capacity (e.g., in a diagnostic lab), the authors must provide some more guidance as to how one can reliably estimate this pre-test probability. Perhaps grouping individuals into risk categories, where they have an interval for their pre-test probability (e.g., $\Pr(D)$ is 0.0001 – 0.001 for low risk), and results are interpreted in this context, rather than a single estimate, might be useful?

3. Finally, as noted above, the stage of infectiousness will impact the sensitivity and specificity of each different type of test (see for example, Boum et al, Lancet Infectious Diseases 2021). Do the authors propose that the sensitivity and specificity be adjusted for each test according to the time that the test was taken (e.g., relative to symptom onset)? If adjusting the narrative away from combining, for example, PCR and serology, then this point may be less important to address. Alternatively, if subtle changes in the sensitivity/specificity due to phase of infectiousness do not substantially impact the estimated posterior probabilities, it may be useful to show this with a sensitivity analysis so that an end-user can understand how precise they have to be with specifying these values.

Decision letter (RSOS-201867.R0)

Dear Dr Wang

The Editors assigned to your paper RSOS-201867 "A Bayesian method for synthesizing multiple diagnostic outcomes of COVID-19" have now received comments from reviewers and would like you to revise the paper in accordance with the reviewer comments and any comments from the Editors. Please note this decision does not guarantee eventual acceptance.

We do not generally allow multiple rounds of revision so we urge you to make every effort to fully address all of the comments at this stage. If deemed necessary by the Editors, your

manuscript will be sent back to one or more of the original reviewers for assessment. If the original reviewers are not available, we may invite new reviewers.

Please submit your revised manuscript and required files (see below) no later than 21 days from today's (ie 13-Apr-2021) date. Note: the ScholarOne system will 'lock' if submission of the revision is attempted 21 or more days after the deadline. If you do not think you will be able to meet this deadline please contact the editorial office immediately.

on behalf of Professor Joshua Ross (Associate Editor) and Mark Chaplain (Subject Editor)
openscience@royalsociety.org

Reviewer comments to Author:
Reviewer: 1
Comments to the Author(s)
Please see uploaded pdf for specific comments.

My key points are:

Firstly, you make a comparison between your method and the use of a single test to make a diagnosis. I do not think this is a fair comparison to make as it confounds two things – firstly the use of multiple tests to make a diagnosis, which even using a completely naïve method would be expected to improve overall diagnostic accuracy, and the benefit of using your method of weighting each test in a probabilistic framework according to its accuracy. What a 'naïve' clinician would usually do would be to look at the outcomes of the multiple tests that they have taken and make a decision based on the proportion of positive tests among those. Again, using a cut-off of 0.5 to determine a positive diagnosis, I think that this would be a much fairer comparison to make with your method to determine its overall benefit.

My second comment is that I think your simulation study is much too restrictive as it is. In the literature, there is usually a great deal of uncertainty as to the exact values of specificity and sensitivity, with ranges of values often given. I think further simulations where you look at the impact of this potential misspecification on your results would make for a better analysis. I think you also need to discuss further the impact of what happens if the underlying disease status of the patient changes across tests. It seems realistic to assume that a patient testing negative on an initial test when not infectious, could then go onto to become infectious by the time of a later test.

There is some alluding to this in the discussion but no mention of the impact of this on your results.

I think you also need to justify your comment about conditional independence between tests. Li and Liu (2019) showed this assumption is not valid for tests on the same biological attribute.

I tried to access your online tool, however the page wouldn't load. I would need to be able to see these as it is a significant part of what your paper is proposing as novel.

Many previous studies have conducted Bayesian analysis of multiple tests and I think a greater reference to this literature and how your method is distinct would emphasise its novelty (e.g. Umemneku Chikere et al (2019); Berkvens et al (2006); Dendukuri (2004)). Is there a methodological novelty or is the novelty that you are applying this method to COVID-19?

References:

Berkvens, Dirk*; Speybroeck, Niko*; Praet, Nicolas*; Adel, Amel†; Lesaffre, Emmanuel†
Estimating Disease Prevalence in a Bayesian Framework Using Probabilistic Constraints,
Epidemiology: March 2006 - Volume 17 - Issue 2 - p 145-153
doi: 10.1097/01.ede.0000198422.64801.8d

Dendukuri, N. and Joseph, L. (2001), Bayesian Approaches to Modeling the Conditional Dependence Between Multiple Diagnostic Tests. *Biometrics*, 57: 158-167.
<https://doi.org/10.1111/j.0006-341X.2001.00158.x>

Li, T., and Liu, P. (2019). Comparison of two Bayesian methods in evaluation of the absence of the gold standard diagnostic tests. *Biomed. Res. Int.* 2019:1374748. doi: 10.1155/2019/1374748

Umemneku Chikere CM, Wilson K, Graziadio S, Vale L, Allen AJ (2019) Diagnostic test evaluation methodology: A systematic review of methods employed to evaluate diagnostic tests in the absence of gold standard – An update. *PLOS ONE* 14(10): e0223832.
<https://doi.org/10.1371/journal.pone.0223832>

Reviewer: 2

Comments to the Author(s)

The authors have used conditional probabilities to estimate the probability that an individual is infected given the results from multiple tests, accounting for the sensitivity and specificity of each test, and the pre-test probability of infection.

The authors have pitched their method as a means to improve diagnostic accuracy to aid with the control of outbreaks, particularly in the context of COVID-19. They suggest that clinicians could use the associated web tool to provide a more accurate estimate of infection status to patients.

There are some disconnects between the theory – which is sound – and how it might be implemented and utilised in practice:

1. The narrative refers to the need for “timely diagnosis” to contain outbreaks, though the example throughout refers to serological testing as one of the three possible tests. The authors themselves note that serological tests provide an accurate indication of prior infection at later stages of infection, which is not consistent with “timely” diagnosis. Particularly for a pathogen such as SARS-CoV-2, where a substantial portion of transmission is pre-symptomatic. Similarly, CT scans are unlikely to be routine for all potential or suspected cases, but rather just for those with more severe disease and/or requiring hospitalisation. As noted further below, the

sensitivity and specificity of these tests varies over time, and each would likely be conducted at different times during the infectious profile of an individual (e.g., PCR early, serology later), making them difficult to be combined simply without additional work to determine the sensitivity/specificity specific to the timing of each test. This makes it difficult to see a situation where these different tests are routinely being performed for timely diagnosis of suspected cases, or the outcomes could be simply and reliably combined. Perhaps it would be appropriate to pitch the narrative along the lines of using multiple rapid diagnostic tests, each of which have lower sensitivity and specificity, but results of which could be combined for a more accurate result.

2. The method requires specification of a pre-test probability to calculate the probability of being infected given the various test results. The authors propose that clinicians specify this probability to calculate the probability the individual is/was infected. While public data is available on daily incidence of cases, it is not immediately clear how the current prevalence would simply inform this probability (e.g., should this be weighted by the generation interval?). Further, these kinds of high-level data cannot account for individual heterogeneities in risk (e.g., healthcare practitioners treating COVID-19 patients would have a higher pre-test probability than another individual in the same population), and thus how much to adjust the pre-test probability in different contexts. Figure 2 in the submitted manuscript highlights the sensitivity of the outcome to this pre-test probability, particularly in the scenarios where it would be most useful – i.e., where the multiple tests give disparate results – and so accurately choosing this probability is key, but difficult. I would propose that were this approach to be implemented in some capacity (e.g., in a diagnostic lab), the authors must provide some more guidance as to how one can reliably estimate this pre-test probability. Perhaps grouping individuals into risk categories, where they have an interval for their pre-test probability (e.g., $\Pr(D)$ is 0.0001 – 0.001 for low risk), and results are interpreted in this context, rather than a single estimate, might be useful?

3. Finally, as noted above, the stage of infectiousness will impact the sensitivity and specificity of each different type of test (see for example, Boum et al, *Lancet Infectious Diseases* 2021). Do the authors propose that the sensitivity and specificity be adjusted for each test according to the time that the test was taken (e.g., relative to symptom onset)? If adjusting the narrative away from combining, for example, PCR and serology, then this point may be less important to address. Alternatively, if subtle changes in the sensitivity/specificity due to phase of infectiousness do not substantially impact the estimated posterior probabilities, it may be useful to show this with a sensitivity analysis so that an end-user can understand how precise they have to be with specifying these values.

===PREPARING YOUR MANUSCRIPT===

===PREPARING YOUR REVISION IN SCHOLARONE===

<https://royalsociety.org/journals/authors/author-guidelines/#data>. You should ensure that

you cite the dataset in your reference list. If you have deposited data etc in the Dryad repository, please include both the 'For publication' link and 'For review' link at this stage.

Author's Response to Decision Letter for (RSOS-201867.R0)

See Appendix B.

RSOS-201867.R1 (Revision)

Review form: Reviewer 1

Is the manuscript scientifically sound in its present form?

Yes

Are the interpretations and conclusions justified by the results?

Yes

Is the language acceptable?

Yes

Do you have any ethical concerns with this paper?

No

Have you any concerns about statistical analyses in this paper?

No

Recommendation?

Accept as is

Comments to the Author(s)

Thank you for addressing my comments fully. I am now happy with the manuscript to be published as it is.

Review form: Reviewer 2

Is the manuscript scientifically sound in its present form?

Yes

Are the interpretations and conclusions justified by the results?

Yes

Is the language acceptable?

Yes

Do you have any ethical concerns with this paper?

No

Have you any concerns about statistical analyses in this paper?

No

Recommendation?

Accept with minor revision (please list in comments)

Comments to the Author(s)

Many thanks for addressing my previous queries. I have only a few minor comments on the current version (line numbers refer to the track changes version):

L181: An individual can test positive when not infectious (e.g., prolonged shedding). This sentence should remove "infectious" so as to not conflate having detectable virus (if the authors are referring to PCR) with being infectious and able to transmit viable virus. For example, the sentence could read: "For example, a patient that tested negative by PCR when first tested shortly after exposure, may not yet have detectable virus, but could later test positive once viral loads are sufficient incubated.", or similar.

L202: I think the word 'suspicious' here should be 'suspected'.

Decision letter (RSOS-201867.R1)

Dear Dr Wang

On behalf of the Editors, we are pleased to inform you that your Manuscript RSOS-201867.R1 "A Bayesian method for synthesizing multiple diagnostic outcomes of COVID-19 tests" has been accepted for publication in Royal Society Open Science subject to minor revision in accordance with the referees' reports. Please find the referees' comments along with any feedback from the Editors below my signature.

We invite you to respond to the comments and revise your manuscript. Below the referees' and Editors' comments (where applicable) we provide additional requirements. Final acceptance of

your manuscript is dependent on these requirements being met. We provide guidance below to help you prepare your revision.

Please submit your revised manuscript and required files (see below) no later than 7 days from today's (ie 17-Aug-2021) date. Note: the ScholarOne system will 'lock' if submission of the revision is attempted 7 or more days after the deadline. If you do not think you will be able to meet this deadline please contact the editorial office immediately.

on behalf of Professor Joshua Ross (Associate Editor) and Mark Chaplain (Subject Editor)
openscience@royalsociety.org

Reviewer comments to Author:

Reviewer: 1

Comments to the Author(s)

Thank you for addressing my comments fully. I am now happy with the manuscript to be published as it is.

Reviewer: 2

Comments to the Author(s)

Many thanks for addressing my previous queries. I have only a few minor comments on the current version (line numbers refer to the track changes version):

L181: An individual can test positive when not infectious (e.g., prolonged shedding). This sentence should remove "infectious" so as to not conflate having detectable virus (if the authors are referring to PCR) with being infectious and able to transmit viable virus. For example, the sentence could read: "For example, a patient that tested negative by PCR when first tested shortly after exposure, may not yet have detectable virus, but could later test positive once viral loads are sufficient incubated.", or similar.

L202: I think the word 'suspicious' here should be 'suspected'.

===PREPARING YOUR MANUSCRIPT===

===PREPARING YOUR REVISION IN SCHOLARONE===

- Any electronic supplementary material (ESM).
- If you are requesting a discretionary waiver for the article processing charge, the waiver form must be included at this step.
- If you are providing image files for potential cover images, please upload these at this step, and inform the editorial office you have done so. You must hold the copyright to any image provided.
- A copy of your point-by-point response to referees and Editors. This will expedite the preparation of your proof.

- Ensure that your data access statement meets the requirements at <https://royalsociety.org/journals/authors/author-guidelines/#data>. You should ensure that you cite the dataset in your reference list. If you have deposited data etc in the Dryad repository, please only include the 'For publication' link at this stage. You should remove the 'For review' link.
- If you are requesting an article processing charge waiver, you must select the relevant waiver option (if requesting a discretionary waiver, the form should have been uploaded at Step 3 'File upload' above).
- If you have uploaded ESM files, please ensure you follow the guidance at <https://royalsociety.org/journals/authors/author-guidelines/#supplementary-material> to include a suitable title and informative caption. An example of appropriate titling and captioning may be found at https://figshare.com/articles/Table_S2_from_Is_there_a_trade-off_between_peak_performance_and_performance_breadth_across_temperatures_for_aerobic_scope_in_teleost_fishes_/3843624.

Author's Response to Decision Letter for (RSOS-201867.R1)

See Appendix C.

Decision letter (RSOS-201867.R2)

Dear Dr Wang,

I am pleased to inform you that your manuscript entitled "A Bayesian method for synthesizing multiple diagnostic outcomes of COVID-19 tests" is now accepted for publication in Royal Society Open Science.

If you have not already done so, please remember to make any data sets or code libraries 'live' prior to publication, and update any links as needed when you receive a proof to check - for

instance, from a private 'for review' URL to a publicly accessible 'for publication' URL. It is good practice to also add data sets, code and other digital materials to your reference list.

COVID-19 rapid publication process:

We are taking steps to expedite the publication of research relevant to the pandemic. If you wish, you can opt to have your paper published as soon as it is ready, rather than waiting for it to be published the scheduled Wednesday.

This means your paper will not be included in the weekly media round-up which the Society sends to journalists ahead of publication. However, it will still appear in the COVID-19 Publishing Collection which journalists will be directed to each week (<https://royalsocietypublishing.org/topic/special-collections/novel-coronavirus-outbreak>).

If you wish to have your paper considered for immediate publication, or to discuss further, please notify openscience_proofs@royalsociety.org and press@royalsociety.org when you respond to this email.

on behalf of Professor Joshua Ross (Associate Editor) and Mark Chaplain (Subject Editor)
openscience@royalsociety.org

Appendix A**ROYAL SOCIETY
OPEN SCIENCE****A Bayesian method for synthesizing multiple diagnostic
outcomes of COVID-19**

Journal:	Royal Society Open Science
Manuscript ID	RSOS-201867
Article Type:	Research
Date Submitted by the Author:	18-Oct-2020
Complete List of Authors:	Cao, Lirong; The Chinese University of Hong Kong; CUHK Shenzhen Research Institute Zhao, Shi; The Chinese University of Hong Kong; CUHK Shenzhen Research Institute Li, Qi; The Chinese University of Hong Kong; CUHK Shenzhen Research Institute Ling, Lowell; The Chinese University of Hong Kong Wu, William; The Chinese University of Hong Kong Zhang, Lin; The Chinese University of Hong Kong Lou, Jingzhi; The Chinese University of Hong Kong Chong, Ka Chun; The Chinese University of Hong Kong; CUHK Shenzhen Research Institute Chen, Zigui; The Chinese University of Hong Kong Lai-yi Wong, Eliza; The Chinese University of Hong Kong Zee, Benny CY; The Chinese University of Hong Kong; CUHK Shenzhen Research Institute Chan, Matthew TV; The Chinese University of Hong Kong Chan, Paul K. S.; The Chinese University of Hong Kong Wang, Maggie H; The Chinese University of Hong Kong; CUHK Shenzhen Research Institute
Subject:	Statistics < MATHEMATICS
Keywords:	COVID-19, SARS-CoV-2, RT-PCR, chest CT, serological tests, multiple tests integration
Subject Category:	Mathematics

**Author-supplied statements**

Relevant information will appear here if provided.

***Ethics***

*Does your article include research that required ethical approval or permits?:*

This article does not present research with ethical considerations

*Statement (if applicable):*

CUST_IF_YES_ETHICS :No data available.

***Data***

*It is a condition of publication that data, code and materials supporting your paper are made publicly*
*available. Does your paper present new data?:*

My paper has no data

*Statement (if applicable):*

CUST_IF_YES_DATA :No data available.

***Conflict of interest***

I/We declare a competing interest

*Statement (if applicable):*

MHW is a shareholder of Beth Bioinformatics Co., Ltd. BCYZ is a shareholder of Beth Bioinformatics
Co., Ltd and Health View Bioanalytics Ltd. Other authors declared no competing interests.

***Authors' contributions***

This paper has multiple authors and our individual contributions were as below

*Statement (if applicable):*

MHW conceived the study. LC conducted the literature review. SZ and MHW carried out the analysis.

LC, SZ, QL and MHW discussed the results. LC and SZ drafted the first manuscript. SZ and QL

developed the online tool. All authors critically read and revised the manuscript and gave final

approval for publication.

**1 Title:** A Bayesian method for synthesizing multiple diagnostic outcomes of COVID-19

**2**

**3** Lirong Cao^{1,2,+}, Shi Zhao^{1,2,+}, Qi Li^{1,2,+}, Lowell Ling³, William KK Wu³, Lin Zhang³, Jingzhi
**4** Lou¹, Marc KC Chong^{1,2}, Zigui Chen³, Eliza Lai-yi Wong¹, Benny CY Zee^{1,2}, Matthew TV
**5** Chan³, Paul KS Chan^{4,5,6}, and Maggie H Wang^{1,2,*}

6
7 1 JC School of Public Health and Primary Care, The Chinese University of Hong Kong, Hong
21
22 Kong, China
23

8
9 2 CUHK Shenzhen Research Institute, Shenzhen, China
24

10
11 3 Department of Anaesthesia and Intensive Care, The Chinese University of Hong Kong, Hong
25
26 Kong SAR, China
27

12
13 4 Department of Microbiology, The Chinese University of Hong Kong, Hong Kong SAR, China
28
29

14
15 5 Stanley Ho Centre for Emerging Infectious Diseases, The Chinese University of Hong Kong,
30
31 Hong Kong SAR, China
32
33

6 Li Ka Shing Institute of Health Sciences, The Chinese University of Hong Kong, Hong Kong
SAR, China

+ Joint first authors.

* Correspondence to: maggiew@cuhk.edu.hk (MHW)

Email addresses of all authors
52
53

22 LC: caolr@link.cuhk.edu.hk
54
55

23 SZ: zhaoshi.cmsa@gmail.com
56
57

24 QL: liqi.cuhk@gmail.com
58
59
60

LL: lowell.ling@cuhk.edu.hk
WKKW: wukakei@cuhk.edu.hk
LZ: linzhang@cuhk.edu.hk
JL: 1156197403@qq.com
MKCC: marc@cuhk.edu.hk
ZC: zigui.chen@cuhk.edu.hk
ELYW: lywong@cuhk.edu.hk
BCYZ: bzee@cuhk.edu.hk
MTVC: mtvchan@cuhk.edu.hk
PKSC: paulkschan@cuhk.edu.hk
MHW: maggiew@cuhk.edu.hk

Abstract

The novel coronavirus disease 2019 (COVID-19) has spread worldwide and threatened human
life. Timely diagnosis is needed to contain the spread of SARS-CoV-2 infections. Diagnostic
tests for COVID-19 have varying sensitivity and specificity, and the false negative results would
have substantial consequences to patient treatment and pandemic control. To detect all suspected
infections, multiple testing is widely used. However, it may be difficult to build an assertion
when the testing results are inconsistent. Considering the situation ~~when there are~~ ^{is} more than one
diagnostic outcomes ^{each} for ~~one~~ ^{where} subject, we proposed a Bayesian probabilistic framework based on
the sensitivity and specificity of each diagnostic method to synthesize a posterior probability of
being infected by SARS-CoV-2. We demonstrated that the synthesized posterior outcome
outperformed each individual testing outcome. A user-friendly web application was developed to
implement our analytic framework with free-access via <http://39.99.171.158:8080/COVID-19/>.
The web application enables real-time display of the integrated outcome incorporating two or
more tests and calculated based on Bayesian posterior probability. A simulation-based
assessment demonstrated higher accuracy and precision of the Bayesian probabilistic model
compared to single-test outcome. The online tool developed in this study can assist physicians in
making clinical evaluations by effectively integrating multiple COVID-19 tests.

**Keywords:** COVID-19, SARS-CoV-2, RT-PCR, chest CT, serological tests, multiple tests
integration

**Background**

The ongoing COVID-19 pandemic is causing substantial morbidity and mortality globally [1].

Timely diagnosis is of importance to control outbreaks, especially in the absence of specific

treatments or vaccines. To date, the COVID-19 is commonly diagnosed by the detection of

unique sequences of SARS-CoV-2 RNA using the Nucleic Acid Amplification Tests (NAAT),

e.g., real-time reverse transcription polymerase chain reaction (RT-PCR) [2]. However,

suboptimal sample collection, storage and transportation may lead to false negative results. The

sensitivity of laboratory-based molecular testing is largely dependent on the types of specimen

and the time of collection from onset of illness [3], which also leads to a large range of

sensitivity for RT-PCR in previous studies, between 46% and 92% [4-8].

As the RT-PCR test might fall short of testing capacity and timeliness in some regions, recent

studies proposed that chest computed tomography (CT) scans could be included as a supplement
diagnostic tool if there is clinical symptom, epidemiological characteristic, and imaging

characteristics of viral pneumonia that are compatible with COVID-19 infection in epidemic

areas [8]. Although the use of chest CT may be useful, the specificity is low due to the absence

of pathognomonic CT features, which is even lower than 50% according to earlier research

findings [9]. In addition, serological tests are also recommended as a supplement for nucleic acid

detection, because the antibody-based methods are relatively cheap, easy to operate and have

lower technical requirements [10]. Since the detection of antibodies against SARS-CoV-2 is

more accurate in the middle to later stages of COVID-19, antibody tests are primarily used to

determine whether a person has been previously infected. In many prior reports, the sensitivity

for combined IgM and IgG detection is higher than 71% and the specificity is higher than 90%

[11-16].

None of the commonly-used diagnostic test, e.g., RT-PCR, chest CT and serological tests, alone
is sufficiently accurate to provide absolute diagnostic certainty [10, 17]. In view of advantages
and shortages of each detection method, parallel or serial tests are recommended in the clinic and
the results are cross-referenced to improve diagnostic yield [18]. However, clinicians will face
immense difficulty diagnosing COVID-19 when the test results are inconsistent. In this study, we
provided a statistical method to synthesize multiple diagnostic outcomes and developed an online
tool to evaluate the probability of an individual being infected by SARS-CoV-2. The online tool
can be applied to assist physicians for diagnosis confirmation of COVID-19.

Would benefit from more detail.

**Methods**

**Bayesian probabilistic framework** With the knowledge of the sensitivity and specificity of
each COVID-19 diagnostic test, we constructed a Bayesian probabilistic model to synthesize
multiple testing outcomes for individual subject. We calculated the posterior probability of
having COVID-19 based on the information and the outcomes of more than one test. Although
the outcomes of different tests may be correlated, they are conditionally independent from each
other with the disease status fixed (knowingly or unknowingly). Thus, the Bayesian probabilistic
framework is applicable to infer the probability of disease status with correlated testing
outcomes. *Needs justification.*

We considered that one individual subject receives M diagnostic tests, where M is an integer and
$M > 1$. The testing outcome, i.e., positive or negative, is denoted by T_i for the i -th test. We
defined T_i as a binary outcome that is 1 for positive testing outcome and 0 otherwise. We denoted
the event that 'the individual subject has COVID-19' by D (stands for 'diagnosed' or 'diseased').
For convenience, we also denoted the complement of event D , i.e., 'the individual subject does

not have COVID-19', by N (stands for 'not diagnosed'). Straightforwardly, the summation of the
probabilities (\Pr) of D and N was 1.

The posterior probability of D ~~on the conditions of~~ ^{conditional on} the M testing outcomes is $\Pr(D|T_1, T_2, \dots,$
$T_M)$. Hence, by using the Bayes theorem, the $\Pr(D|T_1, T_2, \dots, T_M)$ can be computed by using

Equation 1. Based on the intuition of Bayes framework, the test with higher sensitivity or
specificity will be assigned with ~~more weights~~ ^{automatically} ~~automatically~~ ^{a higher weight.}

$$\begin{aligned} \Pr(D | T_1, T_2, \dots, T_M) &= \frac{\Pr(T_1, T_2, \dots, T_M | D) \cdot \Pr(D)}{\Pr(T_1, T_2, \dots, T_M)} && \text{Equation 1.} \\ &= \frac{\Pr(T_1, T_2, \dots, T_M | D) \cdot \Pr(D)}{\Pr(T_1, T_2, \dots, T_M | D) \cdot \Pr(D) + \Pr(T_1, T_2, \dots, T_M | N) \cdot \Pr(N)} \\ &= \frac{\Pr(D) \cdot \prod \Pr(T_i | D)}{\Pr(D) \cdot \prod \Pr(T_i | D) + \Pr(N) \cdot \prod \Pr(T_i | N)}. \end{aligned}$$

Here, $\Pr(T_i = 1|D)$ ^{is} ~~was~~ the sensitivity of the i -th test, and $\Pr(T_i = 0|D)$ ^{is} ~~was~~ $(1 - \text{sensitivity})$.

The $\Pr(T_i = 0|N)$ ^{is} ~~was~~ the specificity of the i -th test, and $\Pr(T_i = 1|N)$ ^{is} ~~was~~ $(1 - \text{specificity})$. The

$\Pr(D)$ indicated the pre-test probability of having COVID-19 for an individual who receives

tests. An alternative interpretation of $\Pr(D)$ is the prevalence of ~~the~~ COVID-19 among the

testing subjects if there is no contact history or symptom. The prevalence among testing

populations is available for each region in "Our World in Data" website [19]. The testing

performance of each COVID-19 diagnostic test can be summarized from ~~the~~ existing literatures ~~.~~

or obtained from the clinical evaluations of commercial testing kits. Hence, $\Pr(D|T_1, T_2, \dots, T_M)$

is computable.

**COVID-19 Diagnostic Assessment Tool** The theoretical framework can be conducted to

integrate multiple diagnostic test results for COVID-19. To facilitate public use, we set up an

open-access user-friendly online application to our framework by web development languages

including HTML, CSS and JavaScript. The application is available at

<http://39.99.171.158:8080/COVID-19/>. The user interface was designed to be intuitive and only

^{currently}
^{made to access.}

has English version ^{can} ~~now~~, see Figure 1. ^{The} “Setup testing information and outcomes” section offers
4 ^{the option}
users to input included tests, sensitivity and specificity, and testing outcome (positive or
negative). Test 1, Test 2 and Test 3 can be RT-PCR, chest CT and antibody test, respectively. Or
it can be testing results from different samples in serial testing. After typing the pre-test
probability for someone with suspected exposure, “Results” section will show the posterior
probability of the subject being infected with COVID-19 according to the above settings. A
detailed user manual is also available in the web application.

**Performance evaluation for the Bayesian method** We further simulated disease diagnosis
according to the posterior probability. The $\Pr(D|T_1, T_2, \dots, T_M) > 0.5$ ^{What's the justification?} was considered as positive
cases for COVID-19. The tested population set in the simulation modelling was one million. The
performance of combined method by integrating multiple testing outcomes was compared with
that of single test by accuracy and precision. The accuracy and precision can be respectively
computed in Equation 2 and Equation 3.

$$33$$

$$34$$

$$35 \text{ Accuracy} = \frac{(TP + TN)}{(TP + TN + FP + FN)} \quad \text{Equation 2.}$$

$$36$$

$$37$$

$$38$$

$$39 \text{ Precision} = \frac{TP}{(TP + FP)} \quad \text{Equation 3.}$$

$$40$$

$$41$$

Here, TP = number of true positive subjects, FP = number of false positive subjects, TN =
number of true negative subjects, and FN = number of false negative subjects. All analysis was
conducted in **R** statistical software (version 3.5.1) [20].

54 140 **Results**

57 141 In Figure 2, we demonstrated the relationship between the probability of having COVID-19 and
58
59
60 142 the prevalence of COVID-19 among testing subjects under a two-test scenario as an example.

The sensitivities were assumed to be at 95% and 80%, and the specificity were assumed to be at
80% and 50%, for test #1 with testing outcome T_1 and test #2 with testing outcome T_2 ,
respectively. Four combinations of the testing outcomes were considered including ($T_1 =$
positive, $T_2 =$ positive) in Fig 1A, ($T_1 =$ positive, $T_2 =$ negative) in Fig 1B, ($T_1 =$ negative, $T_2 =$
positive) in Fig 1C, and ($T_1 =$ negative, $T_2 =$ negative) in Fig 1D. The online tool is also available
for three diagnostic tests synthetization and serial testing synthetization
(<http://39.99.171.158:8080/COVID-19/>).

For comparison purposes, we reported the performance of a single test and the Bayesian *Is this a*
probabilistic model for multiple tests in COVID-19 diagnosis, given a pre-test probability from *fair*
0.001% to 25%, sensitivities and specificities for both T_1 and T_2 from 55% to 100% in Figure 3. *comparison*
T_1 and T_2 could be two types of tests, or repeated tests for different samples. The mean accuracy *?*
for the Bayesian probabilistic model was 88%, which was higher than 81% for T_1 and 74% for
T_2 . The mean precision was also largest for the Bayesian probabilistic model (51%) compared to
T_1 (43%) and T_2 (36%) alone. *Is this due to the Bayesian*
*model or because of multiple*
*tests?*

40 158 Discussion

[revised manuscript text omitted]

**Abbreviations**

Not applicable.

**Declarations**

**Ethics**

Neither ethical approval nor individual consent is applicable.

**Data accessibility**

No real-world data is used in this study. The code supporting this article have been uploaded as
part of the supplementary material.

**Authors' Contributions**

MHW conceived the study. LC conducted the literature review. SZ and MHW carried out the
analysis. LC, SZ, QL and MHW discussed the results. LC and SZ drafted the first manuscript.
SZ and QL developed the online tool. All authors critically read and revised the manuscript and
gave final approval for publication.

**Competing interests**

MHW is a shareholder of Beth Bioinformatics Co., Ltd. BCYZ is a shareholder of Beth
Bioinformatics Co., Ltd and Health View Bioanalytics Ltd. Other authors declared no competing
interests.

**Funding**

This work was supported by CUHK direct grant [grant number 4054524], the Health and
Medical Research Fund (HMRF) Commissioned Research on COVID-19 and [grant number
INF-CUHK-1] of Hong Kong SAR, China, and partially supported by the National Natural
Science Foundation of China (NSFC) [grant number 31871340]. The funding agencies had no
role in the design and conduct of the study; collection, management, analysis, and interpretation

of the data; preparation or approval of the manuscript; or decision to submit the manuscript for
publication.

**Acknowledgements**

None.

References

- WHO Coronavirus Disease (COVID-19) Dashboard. 2020 [cited 2020; Available from: <https://covid19.who.int/>]
- Beeching, N. J., Fletcher, T. E., Beadsworth, M. B. J. 2020 Covid-19: testing times. *Bmj.* **369**, m1403. (doi:10.1136/bmj.m1403)
- Tang, Y. W., Schmitz, J. E., Persing, D. H., Stratton, C. W. 2020 Laboratory Diagnosis of COVID-19: Current Issues and Challenges. *J Clin Microbiol.* **58**, (doi:10.1128/jcm.00512-20)
- Pasomsub, E., Watcharananan, S. P., Boonyawat, K., Janchompoo, P., Wongtabtim, G., Sukswan, W., Sungkanuparph, S., Phuphuakrat, A. 2020 Saliva sample as a non-invasive specimen for the diagnosis of coronavirus disease 2019: a cross-sectional study. *Clin Microbiol Infect.* (doi:10.1016/j.cmi.2020.05.001)
- Wong, H. Y. F., Lam, H. Y. S., Fong, A. H., Leung, S. T., Chin, T. W., Lo, C. S. Y., Lui, M. M., Lee, J. C. Y., Chiu, K. W., Chung, T., *et al.* 2019 Frequency and Distribution of Chest Radiographic Findings in COVID-19 Positive Patients. *Radiology.* 201160. (doi:10.1148/radiol.2020201160)
- He, J. L., Luo, L., Luo, Z. D., Lyu, J. X., Ng, M. Y., Shen, X. P., Wen, Z. 2020 Diagnostic performance between CT and initial real-time RT-PCR for clinically suspected 2019 coronavirus disease (COVID-19) patients outside Wuhan, China. *Respir Med.* **168**, 105980. (doi:10.1016/j.rmed.2020.105980)
- Mei, X., Lee, H. C., Diao, K. Y., Huang, M., Lin, B., Liu, C., Xie, Z., Ma, Y., Robson, P. M., Chung, M., *et al.* 2020 Artificial intelligence-enabled rapid diagnosis of patients with COVID-19. *Nat Med.* (doi:10.1038/s41591-020-0931-3)
- Ai, T., Yang, Z., Hou, H., Zhan, C., Chen, C., Lv, W., Tao, Q., Sun, Z., Xia, L. 2020 Correlation of Chest CT and RT-PCR Testing in Coronavirus Disease 2019 (COVID-19) in China: A Report of 1014 Cases. *Radiology.* 200642. (doi:10.1148/radiol.2020200642)
- Bai, H. X., Hsieh, B., Xiong, Z., Halsey, K., Choi, J. W., Tran, T. M. L., Pan, I., Shi, L. B., Wang, D. C., Mei, J., *et al.* 2020 Performance of radiologists in differentiating COVID-19 from viral pneumonia on chest CT. *Radiology.* 200823. (doi:10.1148/radiol.2020200823)
- Li, C., Ren, L. 2020 Recent progress on the diagnosis of 2019 Novel Coronavirus. *Transbound Emerg Dis.* (doi:10.1111/tbed.13620)
- Lou, B., Li, T. D., Zheng, S. F., Su, Y. Y., Li, Z. Y., Liu, W., Yu, F., Ge, S. X., Zou, Q. D., Yuan, Q., *et al.* 2020 Serology characteristics of SARS-CoV-2 infection since exposure and post symptom onset. *Eur Respir J.* (doi:10.1183/13993003.00763-2020)
- Dellière, S., Salmona, M., Minier, M., Gabassi, A., Alanio, A., Le Goff, J., Delaugerre, C., Chaix, M. L. 2020 Evaluation of COVID-19 IgG/IgM Rapid Test from Orient Gene Biotech. *J Clin Microbiol.* (doi:10.1128/jcm.01233-20)
- Li, Z., Yi, Y., Luo, X., Xiong, N., Liu, Y., Li, S., Sun, R., Wang, Y., Hu, B., Chen, W., *et al.* 2020 Development and clinical application of a rapid IgM-IgG combined antibody test for SARS-CoV-2 infection diagnosis. *J Med Virol.* (doi:10.1002/jmv.25727)
- Cai, X. F., Chen, J., Hu, J. L., Long, Q. X., Deng, H. J., Fan, K., Liao, P., Liu, B. Z., Wu, G. C., Chen, Y. K., *et al.* 2020 A Peptide-based Magnetic Chemiluminescence Enzyme Immunoassay for Serological Diagnosis of Coronavirus Disease 2019 (COVID-19). *J Infect Dis.* (doi:10.1093/infdis/jiaa243)
- Shen, B., Zheng, Y., Zhang, X., Zhang, W., Wang, D., Jin, J., Lin, R., Zhang, Y., Zhu, G., Zhu, H., *et al.* 2020 Clinical evaluation of a rapid colloidal gold immunochromatography assay for SARS-Cov-2 IgM/IgG. *Am J Transl Res.* **12**, 1348-1354.

16 Choe, J. Y., Kim, J. W., Kwon, H. H., Hong, H. L., Jung, C. Y., Jeon, C. H., Park, E. J., Kim, S. K.
2020 Diagnostic performance of immunochromatography assay for rapid detection of IgM and
IgG in coronavirus disease 2019. *J Med Virol.* (doi:10.1002/jmv.26060)
17 Xu, B., Xing, Y., Peng, J., Zheng, Z., Tang, W., Sun, Y., Xu, C., Peng, F. 2020 Chest CT for
detecting COVID-19: a systematic review and meta-analysis of diagnostic accuracy. *Eur Radiol.*
1-8. (doi:10.1007/s00330-020-06934-2)
18 Wang, Y., Hou, H., Wang, W., Wang, W. 2020 Combination of CT and RT-PCR in the screening
or diagnosis of COVID-19. *J Glob Health.* **10**, 010347. (doi:10.7189/jogh.10.010347)
19 Data, O. W. i. COVID-19: Daily tests vs. Daily new confirmed cases per million. 2020 [cited
2020; Available from: <https://ourworldindata.org/grapher/covid-19-daily-tests-vs-daily-new-confirmed-cases-per-million?xScale=linear>
20 Team, R. C. 2013 R: A language and environment for statistical computing.
21 Younes, N., Al-Sadeq, D. W., Al-Jighefee, H., Younes, S., Al-Jamal, O., Daas, H. I., Yassine, H.
312 M., Nasrallah, G. K. 2020 Challenges in Laboratory Diagnosis of the Novel Coronavirus SARS-
313 CoV-2. *Viruses.* **12**, (doi:10.3390/v12060582)
22 Lin, C., Xiang, J., Yan, M., Li, H., Huang, S., Shen, C. 2020 Comparison of throat swabs and
sputum specimens for viral nucleic acid detection in 52 cases of novel coronavirus (SARS-Cov-
2)-infected pneumonia (COVID-19). *Clin Chem Lab Med.* **58**, 1089-1094. (doi:10.1515/cclm-
2020-0187)
23 Wu, J., Liu, J., Li, S., Peng, Z., Xiao, Z., Wang, X., Yan, R., Luo, J. 2020 Detection and analysis of
nucleic acid in various biological samples of COVID-19 patients. *Travel Med Infect Dis.* 101673.
(doi:10.1016/j.tmaid.2020.101673)
24 Xie, X., Zhong, Z., Zhao, W., Zheng, C., Wang, F., Liu, J. 2020 Chest CT for Typical Coronavirus
Disease 2019 (COVID-19) Pneumonia: Relationship to Negative RT-PCR Testing. *Radiology.* **296**,
E41-e45. (doi:10.1148/radiol.2020200343)
25 Adams, H. J. A., Kwee, T. C., Yakar, D., Hope, M. D., Kwee, R. M. 2020 Systematic Review and
Meta-Analysis on the Value of Chest CT in the Diagnosis of Coronavirus Disease (COVID-19): Sol
Scientiae, Illustra Nos. *AJR Am J Roentgenol.* 1-9. (doi:10.2214/ajr.20.23391)
26 Woloshin, S., Patel, N., Kesselheim, A. S. 2020 False Negative Tests for SARS-CoV-2 Infection -
Challenges and Implications. *N Engl J Med.* (doi:10.1056/NEJMp2015897)

COVID19 Diagnostic Assessment Tool

Setup testing information and outcomes (Help)(Example)

Test #1: RT-PCR test	<input type="text" value="included"/>	Sensitivity (#1)	<input type="text" value="0.75"/>	Specificity (#1)	<input type="text" value="0.95"/>	Outcome (#1)	<input type="text" value="positive"/>
Test #2: Chest CT test	<input type="text" value="included"/>	Sensitivity (#2)	<input type="text" value="0.94"/>	Specificity (#2)	<input type="text" value="0.37"/>	Outcome (#2)	<input type="text" value="positive"/>
Test #3: Antibody test	<input type="text" value="included"/>	Sensitivity (#3)	<input type="text" value="0.65"/>	Specificity (#3)	<input type="text" value="0.98"/>	Outcome (#3)	<input type="text" value="positive"/>

Setup pre-test probability for someone with suspected exposure

Pre-test probability

Results

You have selected 3 test(s).

You have set the pre-test probability at 0.5 in the subject population.

The posterior probability of the subject being infected with COVID-19 is 0.999.

330
331
332

Figure 1. User interface for COVID-19 Diagnostic Assessment Tool.

Figure 2. The relationship between $\Pr(D)$ and $\Pr(D|T_1, T_2)$. The sensitivities are assumed at 95% and 80%, and the specificity are assumed at 80% and 50%, for test #1 and test #2 respectively.

Figure 3. Violin plot of the accuracy and precision for test #1, test #2 and the Bayesian probabilistic model. The pre-test probability is assumed from 0.001% to 25%, and sensitivities and specificities for both T1 and T2 are assumed from 55% - 100% in Figure 3.

COVID19 Diagnostic Assessment Tool

Setup testing information and outcomes (Help)(Example)

Test #1: RT-PCR test	<input type="text" value="included"/>	Sensitivity (#1)	<input type="text" value="0.75"/>	Specificity (#1)	<input type="text" value="0.95"/>	Outcome (#1)	<input type="text" value="positive"/>
Test #2: Chest CT test	<input type="text" value="included"/>	Sensitivity (#2)	<input type="text" value="0.94"/>	Specificity (#2)	<input type="text" value="0.37"/>	Outcome (#2)	<input type="text" value="positive"/>
Test #3: Antibody test	<input type="text" value="included"/>	Sensitivity (#3)	<input type="text" value="0.65"/>	Specificity (#3)	<input type="text" value="0.98"/>	Outcome (#3)	<input type="text" value="positive"/>

Setup pre-test probability for someone with suspected exposure

Pre-test probability

Results

You have selected 3 test(s).

You have set the pre-test probability at 0.5 in the subject population.

The posterior probability of the subject being infected with COVID-19 is 0.999.

Figure 1. User interface for COVID-19 Diagnostic Assessment Tool.

Figure 2. The relationship between $\text{Pr}(D)$ and $\text{Pr}(D|T_1, T_2)$. The sensitivities are assumed at 95% and 80%, and the specificities are assumed at 80% and 50%, for test #1 and test #2 respectively.

205x146mm (96 x 96 DPI)

Figure 3. Violin plot of the accuracy and precision for test #1, test #2 and the Bayesian probabilistic model. The pre-test probability is assumed from 0.001% to 25%, and sensitivities and specificities for both T1 and T2 are assumed from 55% - 100% in Figure 3.

Appendix B

Dear Editor,

Thanks for handling our manuscript entitled “*A Bayesian method for synthesizing multiple diagnostic outcomes of COVID-19*” (MS ID: **RSOS-201867**). We appreciate your positive decisions and comments from the reviewers. Please find our point-by-point response below.

Regards,

Maggie H Wang, PhD (corresponding author)

Reviewer #1:

(1) *Specific comments in the uploaded pdf.*

Response: Thank you very much for your valuable comments. We have extensively revised and improved the manuscript as suggested.

(2) *The last sentence in the background part would benefit from more details.*

Response: Thanks for your suggestions. We have improved it with more details.

Line 86: “In this study, we provide a Bayesian method to synthesize multiple diagnostic outcomes and develop an online tool to evaluate the probability of an individual being infected by SARS-CoV-2.”

(3) *“Knowingly or unknowingly” in the Bayesian probabilistic framework of the method section needs justification.*

Response: Thank you for your comments. To avoid confusion, we have removed ‘Knowingly or unknowingly’ in the revised manuscript.

(4) *In the Performance evaluation for the Bayesian method part, “ $Pr(D|T1, T2, \dots, TM) > 0.5$ ” and “one million” need justification.*

Response: Thank you for your comments. The cut-off 0.5 is adopted because 0.5 gives a fair ‘guess’ for binary variables. If one event has a probability larger than 0.5 against its counter side, this event is more likely to occur probabilistically. The ‘one million’ is to mimic the city-level population size. Other numbers, e.g., 10M, 100M or 0.1M, will not alter the main results.

(5) *Is there a fair comparison for comparison of a single test and the Bayesian probabilistic model for multiple tests. Is this due to Bayesian model or because of multiple tests?*

Response: This is a very good question. The common approaches to combine multiple tests are based on calculating a summary statistic from the weighted Z-statistics or p-values (Fisher's method). Such integration is based on the level of significance but cannot account for "effect size" in the summary statistics. The Bayesian approach combines the tests outcomes based on individual tests predicted disease probabilities. Furthermore, the Bayesian method incorporates prior information of disease prevalence, which is very important consideration for infectious disease diagnosis. Because of these differences, it is also hard to have a fair comparison of the Bayesian model against the Fisher's method either, as they are completely different in nature. In clinical setting, Fisher's test is not appropriate to apply to make diagnosis decisions, therefore we present the comparison against the single test in application.

(6) *How is this, "middle or late stage" in the discussion section account for?*

Response: Thank you for the comment. We have revised "middle or late stage" to "after 7 days of symptom onset" in the sentence, as follows:

Line 200: "Therefore, serological tests are more sensitive to detect infections after 7 days of symptom onset (Honacker et al 2020). A negative serological test conducted in the early stage of disease onset may not be sufficient to rule out suspicious cases. Comprehensive reference to multiple testing results, especially results from the rapid diagnostic tests with lower sensitivity and specificity, can provide a more accurate result."

Reference: Van Honacker, E., Coorevits, L., Boelens, J., Verhasselt, B., Van Braeckel, E., Bauters, F., De Bus, L., Schelstraete, P., Willems, J., Vandendriessche, S., *et al.* 2020 Sensitivity and specificity of 14 SARS-CoV-2 serological assays and their diagnostic potential in RT-PCR negative COVID-19 infections. *Acta Clin Belg.* 1-6.

(7) *"However, the calculation of the posterior probability estimation will not be affected." Yes, which is why I think more simulations are necessary. There are many other issues with real data.*

Response: Yes, we agree that the real situations are indeed more complicated. We have conducted more simulations and added the results in the revised manuscript, see Figure S1. The pre-test probability is assumed from 0.001% to 5%; sensitivities and specificities for T1 are assumed from 70% - 100% and 80% - 100%, respectively; sensitivities and specificities for T2 are assumed from 90% - 100% and 50% - 100%, respectively. It is obvious that the performance of the Bayesian probabilistic model is much better than any single test. It is very hard to make specific assumptions in the complex scenarios that might affect the performance. Nevertheless, these different issues would only influence the level of improvement of the performance by the Bayesian framework. With our free-access online tool, users can adjust the parameters according to the real situations and obtain the posterior probability estimation conveniently.

Figure S1. Violin plot of the accuracy and precision for test #1, test #2 and the Bayesian probabilistic model.

(8) *You make a comparison between your method and the use of a single test to make a diagnosis. I do not think this is a fair comparison to make as it confounds two things – firstly the use of multiple tests to make a diagnosis, which even using a completely naïve method would be expected to improve overall diagnostic accuracy, and the benefit of using your method of weighting each test in a probabilistic framework according to its accuracy. What a ‘naïve’ clinician would usually do would be to look at the outcomes of the multiple tests that they have taken and make a decision based on the proportion of positive tests among those. Again, using a cut-off of 0.5 to determine a positive diagnosis, I think that this would be a much fairer comparison to make with your method to determine its overall benefit.*

Response: Thanks for the comments. Here is an example to illustrate the difference between the naïve approach and Bayesian framework. Suppose there are only two tests involved and they gave opposite test results, one positive and the other is negative. In this case, using the majority vote method will result in a random assignment – with no additional information added to the “combined” multiple tests, either statistically or biologically. The classical way of meta-analysis integrates multiple tests either based on a test-statistic value or the p-value while both are based on the level of evidence rather than effect size, and this approach is not ideal to handle the situation of opposite test outcomes. Probability method would provide an appropriate handling by utilizing the predicted probabilities. The Bayesian framework further considers the quality of the individual tests in terms of sensitivity and specificity, which provides a holistic estimation of the test outcome considering the test power (sensitivity and specificity) of individual tests.

(9) *I think your simulation study is much too restrictive as it is. In the literature, there is usually a great deal of uncertainty as to the exact values of specificity and sensitivity, with ranges of values often given. I think further simulations where you look at the impact of this potential misspecification on your results would make for a better analysis. I think you also need to discuss further the impact of what happens if the underlying disease status of the*

patient changes across tests. It seems realistic to assume that a patient testing negative on an initial test when not infectious, could then go onto to become infectious by the time of a later test. There is some alluding to this in the discussion but no mention of the impact of this on your results.

Response: Thank you for your comments. Yes, we agree that the sensitivity and specificity might vary. Therefore, in our simulation, the sensitivity and specificity are not exact values but are assumed from 55% - 100%, see last paragraph in Result section. We have also performed more simulations and added the information in the new manuscript, please see comment (7). The results show that multiple tests combined with our diagnostic tool demonstrated improved diagnostic accuracy and precision compared to individual tests. Furthermore, the underlying disease status of the patient may indeed change across tests. We have added the following content about the impact of underlying disease status in the discussion as suggested.

Line 178: “In serial diagnostic tests applied on a same subject, inconsistent testing results might be observed as a result of the change of underlying disease status rather than the power of the tests themselves. For example, a patient tested negative on an initial test when not infectious could then go onto infectious by the time of a later test. In application, clinicians would also need to consider factors such as exposure risk before and between tests to interpret results.”

(10) *I think you also need to justify your comment about conditional independence between tests. Li and Liu (2019) showed this assumption is not valid for tests on the same biological attribute.*

Response: Thanks for pointing it out. We agree with the reviewer as well as the paper mentioned here. We have revised this part as “*Although the outcomes of different tests may be correlated, they are assumed conditionally independent with the disease status fixed when these tests are based on the different biological attributes.*”

(11) *I tried to access your online tool, however the page wouldn't load. I would need to be able to see these as it is a significant part of what your paper is proposing as novel.*

Response: Thanks for the comment. The online tool is available at http://www2.ccrb.cuhk.edu.hk/statgene/COVID_19/. We have revised the link in the new manuscript.

(12) *Many previous studies have conducted Bayesian analysis of multiple tests and I think a greater reference to this literature and how your method is distinct would emphasise its novelty (e.g. Umemneku Chikere et al (2019); Berkvens et al (2006); Dendukuri (2004)). Is there a methodological novelty or is the novelty that you are applying this method to COVID-19?*

Response: Thank you for your comments. We have added the references in the revised manuscript. Based on the classic Bayesian framework, we applied it in the COVID-19 diagnostic scenario and developed a new web application for flexible use. This online tool enables convenient implementation of synthesized testing, especially for countries or regions currently in the COVID-19 pandemic but lack sufficient medical resources. The online tool can help clinicians calculate the risk of suspected cases with COVID-19.

Reviewer #2:

(1) *The narrative refers to the need for “timely diagnosis” to contain outbreaks, though the example throughout refers to serological testing as one of the three possible tests. The authors themselves note that serological tests provide an accurate indication of prior infection at later stages of infection, which is not consistent with “timely” diagnosis. Particularly for a pathogen such as SARS-CoV-2, where a substantial portion of transmission is pre-symptomatic. Similarly, CT scans are unlikely to be routine for all potential or suspected cases, but rather just for those with more severe disease and/or requiring hospitalisation. As noted further below, the sensitivity and specificity of these tests varies over time, and each would likely be conducted at different times during the infectious profile of an individual (e.g., PCR early, serology later), making them difficult to be combined simply without additional work to determine the sensitivity/specificity specific to the timing of each test. This makes it difficult to see a situation where these different tests are routinely being performed for timely diagnosis of suspected cases, or the outcomes could be simply and reliably combined. Perhaps it would be appropriate to pitch the narrative along the lines of using multiple rapid diagnostic tests, each of which have lower sensitivity and specificity, but results of which could be combined for a more accurate result.*

Response: Thank you very much for your valuable comments. We have revised the manuscript as suggested. We removed the word “Timely” before “Diagnosis” to make the application more general. We also added reviewer’s suggestion addressing the potential use in multiple rapid diagnostic tests with lower sensitivity and specificity.

Line 202: “Comprehensive integration of multiple testing results, especially results from the rapid diagnostic tests with lower sensitivity and specificity, can provide a more accurate result.”

(2) *The method requires specification of a pre-test probability to calculate the probability of being infected given the various test results. The authors propose that clinicians specify this probability to calculate the probability the individual is/was infected. While public data is available on daily incidence of cases, it is not immediately clear how the current prevalence would simply inform this probability (e.g., should this be weighted by the generation interval?). Further, these kinds of high-level data cannot account for individual heterogeneities in risk (e.g., healthcare practitioners treating COVID-19 patients would have a higher pre-test probability than another individual in the same population), and thus how much to adjust the pre-test probability in different contexts.*

Figure 2 in the submitted manuscript highlights the sensitivity of the outcome to this pre-test probability, particularly in the scenarios where it would be most useful — i.e., where the multiple tests give disparate results — and so accurately choosing this probability is key, but difficult. I would propose that were this approach to be implemented in some capacity (e.g., in a diagnostic lab), the authors must provide some more guidance as to how one can reliably estimate this pre-test probability. Perhaps grouping individuals into risk categories, where they have an interval for their pre-test probability (e.g., $Pr(D)$ is 0.0001 – 0.001 for

low risk), and results are interpreted in this context, rather than a single estimate, might be useful?

Response: We are grateful for the valuable comment. Yes, we agree that it is very useful to have an interval for the pre-test probability, but it is difficult for us to provide this interval. We have added the following guidance in the revised manuscript.

Line 118: “Additionally, if the testing subjects are at higher exposure risk, like healthcare practitioners or customs officers, the pre-test probability should be adjusted to a higher range and vice versa.

We have also assumed a lower pre-test probability (Pr(D) is 0.001% to 5%) and conducted further simulations to check the performance of the Bayesian model in low-risk populations. The results are included in the revised manuscript, please see Figure S1. The results are consistent with the previous results that the accuracy and precision of the Bayesian model are better than individual tests.

Figure S1. Violin plot of the accuracy and precision for test #1, test #2 and the Bayesian probabilistic model.

(3) Finally, as noted above, the stage of infectiousness will impact the sensitivity and specificity of each different type of test (see for example, Boum et al, *Lancet Infectious Diseases* 2021). Do the authors propose that the sensitivity and specificity be adjusted for each test according to the time that the test was taken (e.g., relative to symptom onset)? If adjusting the narrative away from combining, for example, PCR and serology, then this point may be less important to address. Alternatively, if subtle changes in the sensitivity/specificity due to phase of infectiousness do not substantially impact the estimated posterior probabilities, it may be useful to show this with a sensitivity analysis so that an end-user can understand how precise they have to be with specifying these values.

Response: We are grateful for the valuable comment. We have added the following content in the methods as the reviewer suggested.

Line 130: “Many factors, such as variation of incubation period, severity of disease, and sample quality, might impact the sensitivity and specificity of diagnostic tests (Boum et al). Users are suggested to adjust these parameters accordingly.”

We appreciate the comments from both reviewers. In this manuscript, we developed a web-based analysis tool using Bayesian method to integrate multiple diagnostic tests, the study is motivated and presented in the scenario of COVID-19 diagnosis. In the online calculator, users can input their desired parameters including pre-test probability, sensitivity and specificity and obtain an integrated test outcome. The specific parameters are subject to a number of other variables such as sampling location, tissue, DNA concentration, stage of disease, severity of diseases, manufacturer claimed test power, and many others, and thus, could only be a rough estimate influenced by these observed or unobserved factors.

Reference: Boum, Y., Fai, K. N., Nicolay, B., Mboringong, A. B., Bebell, L. M., Ndifon, M., Abbah, A., Essaka, R., Eteki, L., Luquero, F., *et al.* 2021 Performance and operational feasibility of antigen and antibody rapid diagnostic tests for COVID-19 in symptomatic and asymptomatic patients in Cameroon: a clinical, prospective, diagnostic accuracy study. *Lancet Infect Dis.*

Appendix C

Dear Editor,

Thank you for considering our manuscript entitled “*A Bayesian method for synthesizing multiple diagnostic outcomes of COVID-19 tests*” (MS ID: **RSOS-201867.R2**) for publication in Royal Society Open Science. We are grateful to you and the reviewers for the valuable suggestions provided. Please find our point-by-point response below.

Regards,

Maggie H Wang, PhD (corresponding author)

Reviewer #1:

(1) *Thank you for addressing my comments fully. I am now happy with the manuscript to be published as it is.*

Response: Thank you for the positive comment. We highly appreciate your professional review work on our manuscript.

Reviewer #2:

(1) *L181: An individual can test positive when not infectious (e.g., prolonged shedding). This sentence should remove "infectious" so as to not conflate having detectable virus (if the authors are referring to PCR) with being infectious and able to transmit viable virus. For example, the sentence could read: "For example, a patient that tested negative by PCR when first tested shortly after exposure, may not yet have detectable virus, but could later test positive once viral loads are sufficient incubated.", or similar.*

Response: Thank you very much for your valuable comments. We have revised the manuscript as suggested.

(2) *L202: I think the word 'suspicious' here should be 'suspected'.*

Response: We are grateful for the correction. We have revised ‘suspicious’ to ‘suspected’ in the sentence.